# Near-Optimal Entrywise Sampling for Data Matrices

**Dimitris Achlioptas**
UC Santa Cruz
optas@cs.ucsc.edu

**Zohar Karnin**
Yahoo Labs
zkarnin@ymail.com

**Edo Liberty**
Yahoo Labs
edo.liberty@ymail.com

## Abstract

We consider the problem of selecting non-zero entries of a matrix $A$ in order to produce a sparse sketch of it, $B$, that minimizes $\|A - B\|_2$. For large $m \times n$ matrices, such that $n \gg m$ (for example, representing $n$ observations over $m$ attributes) we give sampling distributions that exhibit four important properties. First, they have closed forms computable from minimal information regarding $A$. Second, they allow sketching of matrices whose non-zeros are presented to the algorithm in arbitrary order as a stream, with $O(1)$ computation per non-zero. Third, the resulting sketch matrices are not only sparse, but their non-zero entries are highly compressible. Lastly, and most importantly, under mild assumptions, our distributions are provably competitive with the optimal offline distribution. Note that the probabilities in the optimal offline distribution may be complex functions of all the entries in the matrix. Therefore, regardless of computational complexity, the optimal distribution might be impossible to compute in the streaming model.

## 1 Introduction

Given an $m \times n$ matrix $A$, it is often desirable to find a sparser matrix $B$ that is a good proxy for $A$. Besides being a natural mathematical question, such sparsification has become a ubiquitous preprocessing step in a number of data analysis operations including approximate eigenvector computations [AM01, AHK06, AM07], semi-definite programming [AHK05, d'A08], and matrix completion problems [CR09, CT10].

A fruitful measure for the approximation of $A$ by $B$ is the spectral norm of $A - B$, where for any matrix $C$ its spectral norm is defined as $\|C\|_2 = \max_{\|x\|_2=1} \|Cx\|_2$. Randomization has been central in the context of matrix approximations and the overall problem is typically cast as follows: given a matrix $A$ and a budget $s$, devise a distribution over matrices $B$ such that the (expected) number of non-zero entries in $B$ is at most $s$ and $\|A - B\|_2$ is as small as possible.

Our work is motivated by big data matrices that are generated by measurement processes. Each of the $n$ matrix columns correspond to an observation of $m$ attributes. Thus, we expect $n \gg m$. Also we expect the total number of non-zero entries in $A$ to exceed available memory. We assume that the original data matrix $A$ is accessed in the streaming model where we know only very basic features of $A$ a priori and the actual non-zero entries are presented to us one at a time in an arbitrary order. The streaming model is especially important for tasks like recommendation engines where user-item preferences become available one by one in an arbitrary order. But, it is also important in cases when $A$ exists in durable storage and random access of its entries is prohibitively expensive.

We establish that for such matrices the following approach gives provably near-optimal sparsification. Assign to each element $A_{ij}$ of the matrix a weight that depends only on the elements in its row $q_{ij} = |A_{ij}|/\|A_{(i)}\|_1$. Take $\rho$ to be an (appropriate) distribution over the rows. Sample $s$ i.i.d. locations from $A$ using the distribution $p_{ij} = \rho_i q_{ij}$. Return $B$ which is the mean of $s$ matrices, each containing a single non zero entry $A_{ij}/p_{ij}$ in the corresponding selected location $(i, j)$.

As we will see, this simple form of the probabilities $p_{ij}$ falls out naturally from generic optimization considerations. The fact that each entry is kept with probability proportional to its magnitude, be-

sides being interesting on its own right, has a remarkably practical implication. Every non-zero in the $i$-th row of $B$ will take the form $k_{ij}(\|A_{(i)}\|_1/s\rho_i)$ where $|k_{ij}|$ is the number of times location $(i, j)$ of $A$ was selected. Note that since we sample with replacement $|k_{ij}|$ may be more than 1 but, typically, $|k_{ij}| \in \{0, 1\}$. The result is a matrix $B$ which is representable in $O(m \log(n) + s \log(n/s))$ bits. This is because there is no reason to store floating point matrix entry values. We use $O(m \log(n))$ bits to store[1] all values $\|A_{(i)}\|_1/s\rho_i$ and $O(s \log(n/s))$ bits to store the non zero index *offsets*. Note that $\sum |k_{ij}| = s$ and that some of the offsets may be zero. In a simple experiment we measured the average number of bits per sample resulting from this approach (total size of the sketch divided by the number of samples $s$). The results were between 5 and 22 bits per sample depending on the matrix and $s$. It is important to note that the number of bits per sample was usually less than even $\log_2(n) + \log_2(m)$, the minimal number of bits required to represent a pair $(i, j)$. Our experiments show a reduction of disc space by a factor of between 2 and 5 relative to the *compressed* size of the file representing the sample matrix $B$ in the standard row-column-value list format.

Another insight of our work is that the distributions we propose are combinations of two L1-based distributions and and which distribution is more dominant depends on the sampling budget. When the number of samples $s$ is small, $\rho_i$ is nearly linear in $\|A_{(i)}\|_1$ resulting in $p_{ij} \propto |A_{ij}|$. However, as the number of samples grows, $\rho_i$ tends towards $\|A_{(i)}\|_1^2$ resulting in $p_{ij} \propto |A_{ij}| \cdot \|A_{(i)}\|_1$, a distribution we refer to as Row-L1 sampling. The dependence of the preferred distribution on the sample budget is also borne out in experiments, with sampling based on appropriately mixed distributions being consistently best. This highlights that the need to adapt the sampling distribution to the sample budget is a genuine phenomenon.

## 2 Measure of Error and Related Work

We measure the difference between $A$ and $B$ with respect to the L2 (spectral) norm as it is highly revealing in the context of data analysis. Let us define a *linear trend* in the data of $A$ as any tendency of the rows to align with a particular unit vector $x$. To examine the presence of such a trend, we need only multiply $A$ with $x$: the $i$th coordinate of $Ax$ is the projection of the $i$th row of $A$ onto $x$. Thus, $\|Ax\|_2$ measures the strength of linear trend $x$ in $A$, and $\|A\|_2$ measures the strongest linear trend in $A$. Thus, minimizing $\|A-B\|_2$ minimizes the strength of the strongest linear trend of $A$ *not captured* by $B$. In contrast, measuring the difference using an entry-wise norm, e.g., the Frobenius norm, can be completely uninformative. This is because the best strategy would be to always pick the largest $s$ matrix entries from $A$, a strategy that can easily be "fooled". As a stark example, when the matrix entries are $A_{ij} \in \{0, 1\}$, the quality of approximation of $A$ by $B$ is *completely independent* of which elements of $A$ we keep. This is clearly bad; as long as $A$ contains even a modicum of structure certain approximations will be far better than others.

By using the spectral norm to measure error we get a natural and sophisticated target: to minimize $\|A-B\|_2$ is to make $E = A-B$ a near-rotation, having only small variations in the amount by which it stretches different vectors. This idea that the error matrix $E$ should be isotropic, thus packing as much Frobenius norm as possible for its L2 norm, motivated the first work on element-wise matrix sampling by Achlioptas and McSherry [AM07]. Concretely, to minimize $\|E\|_2$ it is natural to aim for a matrix $E$ that is both zero-mean, i.e., an unbiased estimator of $A$, and whose entries are formed by sampling the entries of $A$ (and, thus, of $E$) independently. In the work of [AM07], $E$ is a matrix of i.i.d. zero-mean random variables. The study of the spectral characteristics of such matrices goes back all the way to Wigner's famous semi-circle law [Wig58]. Specifically, to bound $\|E\|_2$ in [AM07] a bound due to Alon Krivelevich and Vu [AKV02] was used, a refinement of a bound by Juhász [Juh81] and Füredi and Komlós [FK81]. The most salient feature of that bound is that it depends on the *maximum* entry-wise variance $\sigma^2$ of $A-B$, and therefore the distribution optimizing the bound is the one in which the variance of all entries in $E$ is the same. In turn, this means keeping each entry of $A$ independently with probability $p_{ij} \propto A_{ij}^2$ (up to a small wrinkle discussed below).

Several papers have since analyzed L2-sampling and variants [NDT09, NDT10, DZ11, GT09, AM07]. An inherent difficulty of L2-sampling based strategies is the need for special handling of small entries. This is because when each item $A_{ij}$ is kept with probability $p_{ij} \propto A_{ij}^2$, the result-

ing entry $B_{ij}$ in the sample matrix has magnitude $|A_{ij}/p_{ij}| \propto |A_{ij}|^{-1}$. Thus, if an extremely small element $A_{ij}$ is accidentally picked, the largest entry of the sample matrix "blows up". In [AM07] this was addressed by sampling small entries with probability proportional to $|A_{ij}|$ rather than $A_{ij}^2$. In the work of Gittens and Tropp [GT09], small entries are not handled separately and the bound derived depends on the ratio between the largest and the smallest non-zero magnitude.

Random matrix theory has witnessed dramatic progress in the last few years and [AW02, RV07, Tro12a, Rec11] provide a good overview of the results. This progress motivated Drineas and Zouzias in [DZ11] to revisit L2-sampling using concentration results for *sums* of random matrices [Rec11], as we do here. This is somewhat different from the original setting of [AM07] since now $B$ is not a random matrix with independent entries, but a sum of many single-element independent matrices, each such matrix resulting by choosing a location of $A$ with replacement. Their work improved upon all previous L2-based sampling results and also upon the L1-sampling result of Arora, Hazan and Kale [AHK06], discussed below, while admitting a remarkably compact proof. The issue of small entries was handled in [DZ11] by deterministically discarding all sufficiently small entries, a strategy that gives a strong mathematical guarantee (but see the discussion regarding deterministic truncation in the experimental section).

A completely different tack at the problem, avoiding random matrix theory altogether, was taken by Arora et al. [AHK06]. Their approximation keeps the largest entries in $A$ deterministically (specifically all $A_{ij} \geqslant \varepsilon/\sqrt{n}$ where the threshold $\varepsilon$ needs be known a priori) and randomly rounds the remaining smaller entries to $\text{sign}(A_{ij})\varepsilon/\sqrt{n}$ or 0. They exploit the simple fact $\|A - B\| = \sup_{\|x\|=1, \|y\|=1} x^T(A - B)y$ by noting that, as a scalar quantity, its concentration around its expectation can be established by standard Bernstein-Bennet type inequalities. A union bound then allows them to prove that with high probability, $x^T(A - B)y \leqslant \varepsilon$ for *every* $x$ and $y$. The result of [AHK06] admits a relatively simple proof. However, it also requires a truncation that depends on the desired approximation $\varepsilon$. Rather interestingly, this time the truncation amounts to keeping every entry *larger* than some threshold.

# 3   Our Approach

Following the discussion in Section 2 and in line with previous works, we: (i) measure the quality of $B$ by $\|A - B\|_2$, (ii) sample the entries of $A$ independently, and (iii) require $B$ to be an unbiased estimator of $A$. We are therefore left with the task of determining a good probability distribution $p_{ij}$ from which to sample the entries of $A$ in order to get $B$. As discussed in Section 2 prior art makes heavy use of beautiful results in the theory of random matrices. Specifically, each work proposes a specific sampling distribution and then uses results from random matrix theory to demonstrate that it has good properties. In this work we reverse the approach, aiming for its logical conclusion. We start from a cornerstone result in random matrix theory and work backwards to reverse-engineer near-optimal distributions with respect to the notion of probabilistic deviations captured by the inequality. The inequality we use is the Matrix-Bernstein inequality for sums of independent random matrices (see e.g., [Tro12b], Theorem 1.6). In the following, we often denote $\|A\|_2$ as $\|A\|$ to lighten notation.

**Theorem 3.1** (Matrix Bernstein inequality). *Consider a finite sequence $\{X_i\}$ of i.i.d. random $m \times n$ matrices, where $\mathbb{E}[X_1] = 0$ and $\|X_1\| \leqslant R$. Let $\sigma^2 = \max\left\{\|\mathbb{E}[X_1 X_1^T]\|, \|\mathbb{E}[X_1^T X_1]\|\right\}$.*

*For some fixed $s \geqslant 1$, let $X = (X_1 + \cdots + X_s)/s$. For all $\varepsilon \geqslant 0$,*

$$\Pr[\|X\| \geqslant \varepsilon] \leqslant (m + n) \exp\left(-\frac{s\varepsilon^2}{\sigma^2 + R\varepsilon/3}\right) \ .$$

To get a feeling for our approach, fix any probability distribution $p$ over the non-zero elements of $A$. Let $B$ be a random $m \times n$ matrix with exactly one non-zero element, formed by sampling an element $A_{ij}$ of $A$ according to $p$ and letting $B_{ij} = A_{ij}/p_{ij}$. Observe that for every $(i, j)$, regardless of the choice of $p$, we have $\mathbb{E}[B_{ij}] = A_{ij}$, and thus $B$ is always an unbiased estimator of $A$. Clearly, the same is true if we repeat this $s$ times, taking i.i.d. samples $B_1, \ldots, B_s$, and let our matrix $B$ be their average. With this approach in mind, the goal is now to find a distribution $p$ minimizing $\|E\| = \|A - (B_1 + \cdots + B_s)/s\|$. Writing $sE = (A - B_1) + \cdots + (A - B_s)$ we see that $\|sE\|$ is the operator norm of a sum of i.i.d. zero-mean random matrices $X_i = A - B_i$, i.e., exactly the setting

of Theorem 3.1. The relevant parameters are

$$\sigma^2 = \max\left\{\|\mathbb{E}[(A-B_1)(A-B_1)^T]\|, \|\mathbb{E}[(A-B_1)^T(A-B_1)]\|\right\} \quad (1)$$
$$R = \max\|A - B_1\| \quad \text{over all possible realizations of } B_1 \text{ .} \quad (2)$$

Equations (1) and (2) mark the starting point of our work. Our goal is to find probability distributions over the elements of $A$ that optimize (1) and (2) *simultaneously* with respect to their functional form in Theorem 3.1, thus yielding the strongest possible bound on $\|A - B\|$. A conceptual contribution of our work is the discovery that good distributions *depend* on the sample budget $s$, a fact also borne out in experiments. The fact that minimizing the deviation metric of Theorem 3.1, i.e., $\sigma^2 + R\epsilon/3$, suffices to bring out this dependence can be viewed as testament to the theorem's sharpness.

Theorem 3.1 is stated as a bound on the probability that the norm of the error matrix is greater than some target error $\varepsilon$ given the number of samples $s$. In practice, the target error $\varepsilon$ is typically not known in advance, but rather is the quantity to minimize, given the matrix $A$, the number of samples $s$, and the target confidence $\delta$. Specifically, for any given distribution $p$ on the elements of $A$, define

$$\varepsilon_1(p) = \inf\left\{\varepsilon : (m+n)\exp\left(-\frac{s\varepsilon^2}{\sigma(p)^2 + R(p)\varepsilon/3}\right) \leqslant \delta\right\} \quad . \quad (3)$$

Our goal in the rest of the paper is to seek the distribution $p^*$ minimizing $\varepsilon_1$. Our result is an easily computable distribution $p$ which comes within a factor of 3 of $\varepsilon_1(p^*)$ and, as a result, within a factor of 9 in terms of sample complexity (in practice we expect this to be even smaller, as the factor of 3 comes from consolidating bounds for a number of different worst-case matrices). To put this in perspective note that the definition of $p^*$ does not place *any* restriction either on the access model for $A$ while computing $p^*$, or on the amount of time needed to compute $p^*$. In other words, we are competing against an oracle which in order to determine $p^*$ has *all* of $A$ in its purview at once and can spend an unbounded amount of computation to determine it.

In contrast, the only global information regarding $A$ we require are the *ratios* between the L1 norms of the rows of the matrix. Trivially, the exact L1 norms of the rows (and therefore their ratios) can be computed in a single pass over the matrix, yielding a 2-pass algorithm. Slightly less trivially, standard concentration arguments imply that these ratios can be estimated very well by sampling only a small number of columns. In the setting of data analysis, though, it is in fact reasonable to expect that good estimates of these ratios are available *a priori*. This is because different rows correspond to different attributes and the ratios between the row norms reflect the ratios between the average absolute values of the features. For example, if the matrix corresponds to text documents, knowing the ratios amounts to knowing global word frequencies. Moreover these ratios do not need to be known exactly to apply the algorithm, as even rough estimates of them give highly competitive results. Indeed, even disregarding this issue completely and simply assuming that all ratios equal 1, yields an algorithm that appears quite competitive in practice, as demonstrated by our experiments.

## 4  Data Matrices and Statement of Results

Throughout $A_{(i)}$ and $A^{(j)}$ will denote the $i$-th row and $j$-th column of $A$, respectively. Also, we use the notation $\|A\|_1 = \sum_{i,j}|A_{ij}|$ and $\|A\|_F^2 = \sum_{i,j}A_{ij}^2$. Before we formally state our result we introduce a definition that expresses the class of matrices for which our results hold.

**Definition 4.1.** *An $m \times n$ matrix $A$ is a* Data *matrix if:*

1. $\min_i \|A_{(i)}\|_1 \geqslant \max_j \|A^{(j)}\|_1$.
2. $\|A\|_1^2/\|A\|_2^2 \geqslant 30m$.
3. $m \geqslant 30$.

Regarding Condition 1, recall that we think of $A$ as being generated by a measurement process of a fixed number of attributes (rows), each column corresponding to an observation. As a result, columns have bounded L1 norm, i.e., $\|A^{(j)}\|_1 \leqslant$ constant. While this constant may depend on the type of object and its dimensionality, it is independent of the number of objects. On the other hand, $\|A_{(i)}\|_1$ grows linearly with the number of columns (objects). As a result, we can expect Definition 4.1 to hold for all large enough data sets. Regarding Condition 2, it is easy to verify that

unless the values of the entries of $A$ exhibit unbounded variance as $n$ grows, the ratio $\|A\|_1^2/\|A\|_2^2$ grows as $\Omega(n)$ and Condition 2 follows from $n \gg m$. Condition 3 is trivial. All in all, out of the three conditions the essential one is Condition 1. The other two are merely technical and hold in all non-trivial cases where Condition 1 applies.

One last point is that to apply Theorem 3.1, the entries of $A$ must be sampled *with* replacement. A simple way to achieve this in the streaming model was presented in [DKM06] that uses $O(s)$ operations per matrix element and $O(s)$ active memory. In Section D (see supplementary material) we discuss how to implement sampling with replacement far more efficiently, using $O(\log s)$ active memory, $\tilde{O}(s)$ space, and $O(1)$ operations per element. To simplify the exposition of our algorithm, below, we describe it in the *non-streaming* setting. That is, we assume we know $m$ and $n$ and that we can compute numbers $z_i \propto \|A_{(i)}\|_1$ as well as repeatedly sample entries from the matrix. We stress, however, that these conditions are not required and that the algorithm can be implemented efficiently in the streaming model as discussed in Section D.

---

**Algorithm 1** Construct a sketch $B$ of a data matrix $A$

---

1: **Input:** Data matrix $A \in \mathbb{R}^{m \times n}$, sampling budget $s$, acceptable failure probability $\delta$
2: Set $\rho \leftarrow \textsc{ComputeRowDistribution}(A, s, \delta)$
3: Sample $s$ elements of $A$ with replacement, each $A_{ij}$ having probability $p_{ij} = \rho_i \cdot |A_{ij}|/\|A_{(i)}\|_1$
4: For each sample $\langle i, j, A_{ij}\rangle_\ell$, let $B_\ell$ be the matrix with $B_\ell(i,j) = A_{ij}/p_{ij}$ and zero elsewhere.
5: **Output:** $B = \frac{1}{s}\sum_{\ell=1}^s B_\ell$.

---

6: **function** $\textsc{ComputeRowDistribution}(A, s, \delta)$
7:     Obtain $z$ such that $z_i \propto \|A_{(i)}\|_1$ for $i \in [m]$
8:     Set $\alpha \leftarrow \sqrt{\log((m+n)/\delta)/s}$     and     $\beta \leftarrow \log((m+n)/\delta)/(3s)$
9:     Define $\rho_i(\zeta) = \left(\alpha z_i/2\zeta + \sqrt{(\alpha z_i/2\zeta)^2 + \beta z_i/\zeta}\right)^2$
10:     Find $\zeta_1$ such that $\sum_{i=1}^m \rho_i(\zeta_1) = 1$
11:     **return** $\rho$ such that $\rho_i = \rho_i(\zeta_1)$ for $i \in [m]$

---

Steps 6–11 compute a distribution $\rho$ over the rows. Assuming step 7 can be implemented efficiently (or skipped altogether in the case $z$ are known a priori), we see that the running time of ComputeRowDistribution is independent of $n$. Specifically, finding $\zeta_1$ in step 10 can be done efficiently by binary search because the function $\sum_i \rho_i(\zeta)$ is strictly decreasing in $\zeta$. Conceptually, we see that the probability assigned to each element $A_{ij}$ in Step 3 is simply the probability $\rho_i$ of its row times its intra-row weight $|A_{ij}|/\|A_{(i)}\|_1$.

We are now able to state our main lemma. We defer its proof to Section 5 and subsequent details to appendices (see supplementary material).

**Theorem 4.2.** *If $A$ is a Data matrix per Definition 4.1 and $p$ is the probability distribution defined in Algorithm 1, then $\varepsilon_1(p) \leqslant 3\,\varepsilon_1(p^*)$, where $p^*$ is the minimizer of $\varepsilon_1$.*

To compare our result with previous ones we first define several matrix metrics. We then state the bound implied by Theorem 4.2 on the minimal number of samples $s_0$ needed by our algorithm to achieve an approximation $B$ to the matrix $A$ such that $\|A - B\| \leqslant \varepsilon\|A\|$ with constant probability.

**Stable rank**: Denoted as $\mathrm{sr}$ and defined as $\|A\|_F^2/\|A\|_2^2$. This is a smooth analog for the algebraic rank, always bounded by it from above, and resilient to small perturbations of the matrix. For data matrices we expect it is small, even constant, and that it captures the "inherent dimensionality" of the data.

**Numeric density**: Denoted as $\mathrm{nd}$ and defined as $\|A\|_1^2/\|A\|_F^2$, this is a smooth analog of the number of non-zero entries $\mathrm{nnz}(A)$. For 0-1 matrices it equals $\mathrm{nnz}(A)$, but when there is variance in the magnitude of the entries it is smaller.

**Numeric row density**: Denoted as $\mathrm{nrd}$ and defined as $\sum_i \|A_{(i)}\|_1^2/\|A\|_F^2 \leqslant n$. In practice, it is often close to the average numeric density of a single row, a quantity typically much smaller than $n$.

**Theorem 4.3.** *Let $A$ be a Data Matrix per Definition 4.1 and let $B$ be the matrix returned by Algorithm 1 for $\delta = 1/10$, $\varepsilon > 0$ and any*

$$s \geqslant s_0 = \Theta(\mathrm{nrd} \cdot \mathrm{sr}/\varepsilon^2 \cdot \log n + (\mathrm{sr} \cdot \mathrm{nd}/\varepsilon^2 \cdot \log n)^{1/2}) \ .$$

*With probability at least $9/10$, $\|A - B\| \leqslant \varepsilon\|A\|$.*

The proof of Theorem 4.3 is given in Appendix C (see supplementary material).

The third column of the table below shows the number of samples needed to guarantee that $\|A - B\| \leqslant \varepsilon\|A\|$ occurs with constant probability, in terms of the matrix metrics defined above. The fourth column presents the ratio of the samples needed by previous results divided by the samples needed by our method. (To simplify the expressions, we present the ratio between our bound and [AHK06] only when the result of [AHK06] gives superior bounds to [DZ11], i.e., we always compare our bound to the stronger of the two bounds implied by these works). Holding $\varepsilon$ and the stable rank constant we readily see that our method requires roughly $1/\sqrt{n}$ the samples needed by [AHK06]. In the comparison with [DZ11] we see that the key parameter is the ratio $\mathrm{nrd}/n$, a quantity typically much smaller than 1 for data matrices. As a point of reference for the assumptions, in the experimental Section 6 we provide the values of all relevant matrix metrics for all the real data matrices we worked with, wherein the ratio $\mathrm{nrd}/n$ is typically around $10^{-2}$. By this discussion, one would expect that L2-sampling should fare better than L1-sampling in experiments. As we will see, quite the opposite is true. A potential explanation for this phenomenon is the relative looseness of the bound of [AHK06] for the performance of L1-sampling.

| Citation | Method | Number of samples needed | Improvement ratio of Theorem 4.3 |
|----------|--------|--------------------------|----------------------------------|
| [AM07] | L1, L2 | $\mathrm{sr} \cdot (n/\varepsilon^2) + n \cdot \mathrm{polylog}(n)$ | |
| [DZ11] | L2 | $\mathrm{sr} \cdot (n/\varepsilon^2) \log(n)$ | $\mathrm{nrd}/n + (\sqrt{\mathrm{nd}}/n) \cdot (\varepsilon/\sqrt{\mathrm{sr}\log(n)})$ |
| [AHK06] | L1 | $(\mathrm{nd} \cdot n/\varepsilon^2)^{1/2}$ | $\sqrt{\mathrm{sr} \cdot \log(n)/n}$ |
| This paper | Bernstein | $\mathrm{nrd} \cdot \mathrm{sr}/\varepsilon^2 \cdot \log n +$ $(\mathrm{sr} \cdot \mathrm{nd}/\varepsilon^2 \cdot \log n)^{1/2}$ | |

## 5   Proof outline

We start by iteratively replacing the objective functions (1) and (2) with simpler and simpler functions. Each replacement will incur a (small) loss in accuracy but will bring us closer to a function for which we can give a closed form solution. Recalling the definitions of $\alpha, \beta$ from Algorithm 1 and rewriting the requirement in (3) as a quadratic form in $\varepsilon$ gives $\varepsilon^2 - \varepsilon\beta R - (\alpha\sigma)^2 > 0$. Our first step is to observe that for any $c, d > 0$, the equation $\varepsilon^2 - \varepsilon \cdot c - d = 0$ has one negative and one positive solution and that the latter is at least $(c + \sqrt{d})/\sqrt{2}$ and at most $c + \sqrt{d}$. Therefore, if we define[2] $\varepsilon_2 := \alpha\sigma + \beta R$ we see that $1/\sqrt{2} \leqslant \varepsilon_1/\varepsilon_2 \leqslant 1$.

Our next simplification encompasses Conditions 2, 3 of Definition 4.1. Let $\varepsilon_3 := \alpha\tilde{\sigma} + \beta\tilde{R}$ where

$$\tilde{\sigma}^2 := \max\left\{\max_i \sum_j A_{ij}^2/p_{ij} \ , \ \max_j \sum_i A_{ij}^2/p_{ij}\right\} \quad \text{and} \quad \tilde{R} := \max_{ij} |A_{ij}|/p_{ij} \ .$$

**Lemma 5.1.** *For every matrix $A$ satisfying Conditions 2 and 3 of Definition 4.1, for every probability distribution on the elements of $A$, $|\varepsilon_2/\varepsilon_3 - 1| \leqslant 1/30$.*

Lemma 5.1 is proved in section A (see supplementary material) by showing that $\tilde{\sigma} \approx \sigma$ and $\tilde{R} \approx R$. This allows us to optimize $p$ with respect to $\varepsilon_3$ instead of $\varepsilon_2$. In minimizing $\varepsilon_3$ we see that there is freedom to use different rows to optimize $\tilde{\sigma}$ and $\tilde{R}$. At a cost of a factor of 2, we will couple the two

minimizations by minimizing $\varepsilon_4 = \max\{\varepsilon_5, \varepsilon_6\}$ where

$$\varepsilon_5 := \max_i \left[ \alpha \sqrt{\sum_j \frac{A_{ij}^2}{p_{ij}}} + \beta \max_j \frac{|A_{ij}|}{p_{ij}} \right], \qquad \varepsilon_6 := \max_j \left[ \alpha \sqrt{\sum_i \frac{A_{ij}^2}{p_{ij}}} + \beta \max_i \frac{|A_{ij}|}{p_{ij}} \right] . \quad (4)$$

Note that the maximization of $\tilde{R}$ in $\varepsilon_5$ (and $\varepsilon_6$) is coupled with that of the $\tilde{\sigma}$-related term by constraining the optimization to consider only one row (column) at a time. Clearly, $1 \leqslant \varepsilon_3/\varepsilon_4 \leqslant 2$.

Next we focus on $\varepsilon_5$, the first term in the maximization of $\varepsilon_4$. The following key lemma establishes that for all data matrices satisfying Condition 1 of Definition 4.1, by minimizing $\varepsilon_5$ we also minimize $\varepsilon_4 = \max\{\varepsilon_5, \varepsilon_6\}$.

**Lemma 5.2.** *For every matrix satisfying Condition 1 of Definition 4.1,* $\mathrm{argmin}_p \varepsilon_5 \subseteq \mathrm{argmin}_p \varepsilon_4$.

At this point we can derive in closed form the probability distribution $p$ minimizing $\varepsilon_5$.

**Lemma 5.3.** *The function $\varepsilon_5$ is minimized by $p_{ij} = \rho_i q_{ij}$ where $q_{ij} = |A_{ij}|/\|A_{(i)}\|_1$. To define $\rho_i$ let $z_i \propto \|A_{(i)}\|_1$ and define $\rho_i(\zeta) = \left( \alpha z_i/2\zeta + \sqrt{(\alpha z_i/2\zeta)^2 + \beta z_i/\zeta} \right)^2$. Let $\zeta_1 > 0$ be the unique solution to[3] $\sum_i \rho_i(\zeta_1) = 1$. Let $\rho_i := \rho_i(\zeta_1)$.*

To prove Theorem 4.2 we see that Lemmas 5.2 and 5.3 combined imply that there is an efficient algorithm for minimizing $\varepsilon_4$ for every matrix $A$ satisfying Condition 1 of Definition 4.1. If $A$ also satisfies Conditions 2 and 3 of Definition 4.1, then it is possible to lower and upper bound the ratios $\varepsilon_1/\varepsilon_2$, $\varepsilon_2/\varepsilon_3$ and $\varepsilon_3/\varepsilon_4$. Combined, these bounds guarantee a lower and upper bound for $\varepsilon_1/\varepsilon_4$. In general, if $c \leqslant \varepsilon_4/\varepsilon_1 \leqslant C$ we can conclude that $\varepsilon_1(\arg\min(\varepsilon_4)) \leqslant (C/c)\min(\varepsilon_1)$. Thus, calculating the constants shows $\varepsilon_1(\arg\min(\varepsilon_4)) \leqslant 3\min(\varepsilon_1)$, yielding Theorem 4.3.

## 6    Experiments

We experimented with 4 matrices with different characteristics, summarized in the table below. See Section 4 for the definition of the different characteristics.

| Measure | $m$ | $n$ | $\mathrm{nnz}(A)$ | $\|A\|_1$ | $\|A\|_F$ | $\|A\|_2$ | sr | nd | nrd |
|---|---|---|---|---|---|---|---|---|---|
| Synthetic | 1.0e+2 | 1.0e+4 | 5.0e+5 | 1.8e+7 | 3.2e+4 | 8.7e+3 | 1.3e+1 | 3.1e+5 | 3.2e+3 |
| Enron | 1.3e+4 | 1.8e+5 | 7.2e+5 | 4.0e+9 | 5.8e+6 | 1.0e+6 | 3.2e+1 | 4.9e+5 | 1.5e+3 |
| Images | 5.1e+3 | 4.9e+5 | 2.5e+8 | 6.5e+9 | 2.0e+6 | 1.8e+6 | 1.3e+0 | 1.1e+7 | 2.3e+3 |
| Wikipedia | 4.4e+5 | 3.4e+6 | 5.3e+8 | 5.3e+9 | 7.5e+5 | 1.6e+5 | 2.1e+1 | 5.0e+7 | 1.9e+4 |

**Enron:** Subject lines of emails in the Enron email corpus [Sty11]. Columns correspond to subject lines, rows to words, and entries to tf-idf values. This matrix is extremely sparse to begin with.
**Wikipedia:** Term-document matrix of a fragment of Wikipedia in English. Entries are tf-idf values.
**Images:** A collection of images of buildings from Oxford [PCI$^+$07]. Each column represents the wavelet transform of a single $128 \times 128$ pixel grayscale image.
**Synthetic:** This synthetic matrix simulates a collaborative filtering matrix. Each row corresponds to an item and each column to a user. Each user and each item was first assigned a random latent vector (i.i.d. Gaussian). Each value in the matrix is the dot product of the corresponding latent vectors plus additional Gaussian noise. We simulated the fact that some items are more popular than others by retaining each entry of each item $i$ with probability $1 - i/m$ where $i = 0, \dots, m - 1$.

### 6.1    Sampling techniques and quality measure

The experiments report the accuracy of sampling according to four different distributions. In Figure 6.1, **Bernstein** denotes the distribution of this paper, defined in Lemma 5.3. The **Row-L1** distribution is a simplified version of the Bernstein distribution, where $p_{ij} \propto |A_{ij}| \cdot \|A_{(i)}\|_1$. **L1** and **L2** refer to $p_{ij} \propto |A_{ij}|$ and $p_{ij} \propto |A_{ij}|^2$, respectively, as defined earlier in the paper. The case of **L2**

sampling was split into three sampling methods corresponding to different trimming thresholds. In the method referred to as **L2** no trimming is made and $p_{ij} \propto |A_{ij}|^2$. In the case referred to as **L2 trim 0.1**, $p_{ij} \propto |A_{ij}|^2$ for any entry where $|A_{ij}|^2 > 0.1 \cdot \mathbb{E}_{ij}[|A_{ij}|^2]$ and $p_{ij} = 0$ otherwise. The sampling technique referred to as **L2 trim 0.01** is analogous with threshold $0.01 \cdot \mathbb{E}_{ij}[|A_{ij}|^2]$.

Although to derive our sampling probability distributions we targeted minimizing $\|A - B\|_2$, in experiments it is more informative to consider a more sensitive measure of quality of approximation. The reason is that for a number of values of $s$, the scaling of entries required for $B$ to be an unbiased estimator of $A$, results in $\|A - B\| > \|A\|$ which would suggest that the all zeros matrix is a better sketch for $A$ than the sampled matrix. We will see that this is far from being the case. As a trivial example, consider the possibility $B \approx 10A$. Clearly, $B$ is very informative of $A$ although $\|A - B\| \geqslant 9\|A\|$. To avoid this pitfall, we measure $\|P_k^B A\|_F / \|A_k\|_F$, where $P_k^B$ is the projection on the top $k$ left singular vectors of $B$. Thus, $A_k = P_k^A A$ is the optimal rank $k$ approximation of $A$. Intuitively, this measures how well the top $k$ left singular vectors of $B$ capture $A$, compared to $A$'s own (optimal) top-$k$ left singular vectors. We also compute $\|AQ_k^B\|_F / \|A_k\|_F$ where $Q_k^B$ is the projection on the top $k$ right singular vectors of $A$. Note that, for a given $k$, approximating the row-space is harder than approximating the column-space since it is of dimension $n$ which is significantly larger than $m$, a fact also borne out in the experiments. In the experiments we made sure to choose a sufficiently wide range of sample sizes so that at least the best method for each matrix goes from poor to near-perfect both in approximating the row and the column space. In all cases we report on $k = 20$ which is close to the upper end of what could be efficiently computed on a single machine for matrices of this size. The results for all smaller values of $k$ are qualitatively indistinguishable.

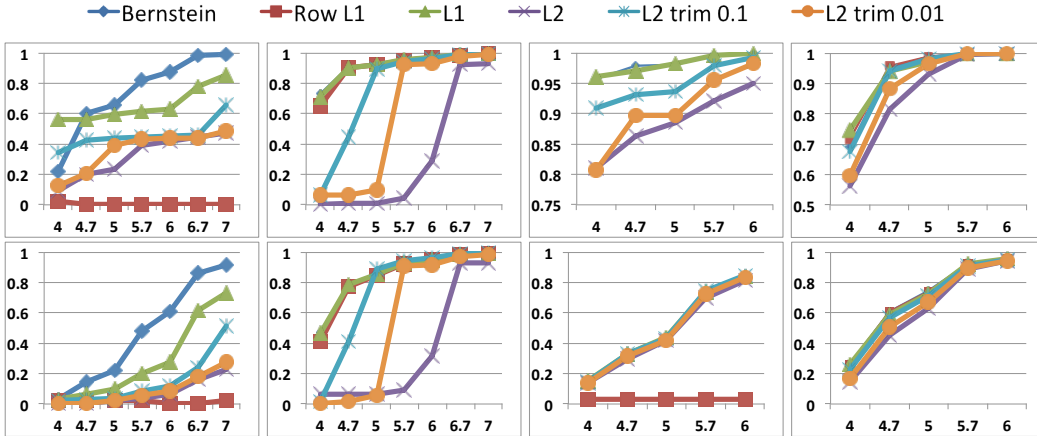

Figure 1: Each vertical pair of plots corresponds to one matrix. Left to right: Wikipedia, Images, Enron, Synthetic . Each top plot shows the quality of the column-space approximation ratio, $\|P_B^k A\|_F / \|A_k\|$, while the bottom plots show the row-space approximation ratio $\|AQ_B^k\|_F / \|A_k\|$. The number of samples $s$ is on the $x$-axis in log scale $x = \log_{10}(s)$.

## 6.2  Insights

The experiments demonstrate three main insights. First and most important, Bernstein-sampling is never worse than any of the other techniques and is often strictly better. A dramatic example of this is the Wikipedia matrix for which it is far superior to all other methods. The second insight is that L1-sampling, i.e., simply taking $p_{ij} = |A_{ij}|/\|A\|_1$, performs rather well in many cases. Hence, if it is impossible to perform more than one pass over the matrix and one can not even obtain an estimate of the ratios of the L1-weights of the rows, L1-sampling seems to be a highly viable option. The third insight is that for L2-sampling, discarding small entries may drastically improve the performance. However, it is not clear which threshold should be chosen in advance. In any case, in all of the example matrices, both L1-sampling and Bernstein-sampling proved to outperform or perform equally to L2-sampling, even with the correct trimming threshold.

## Footnotes

[1]It is harmless to assume any value in the matrix is kept using $O(\log(n))$ bits of precision. Otherwise, truncating the trailing bits can be shown to be negligible.

[2]Here and in the following, to lighten notation, we will omit all arguments, i.e., $p, \sigma(p), R(p)$, from the objective functions $\varepsilon_i$ we seeks to optimize, as they are readily understood from context.

[3]Notice that the function $\sum \rho_i(\zeta)$ is monotonically decreasing for $\zeta > 0$ hence the solution is indeed unique.

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
