[Supplementary Material]

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

# A  Optimizations on the L1 ball

**Lemma A.1.** *For any $x, p \in \mathbb{R}^n$, if $p_i \geqslant 0$ and $\|p\|_1 = 1$, then $\max_k |x_k|/p_k \geqslant \|x\|_1$ and $\sum_k x_k^2/p_k \geqslant \|x\|_1^2$, with equality holding in both cases if and only if $p_k = |x_k|/\|x\|_1$.*

*Proof.* To prove $\max_k |x_k|/p_k \geqslant \|x\|_1$ we note that if $|x_i|/p_i \neq |x_j|/p_j$, then changing $p_i, p_j$ to $p_i', p_j'$ such that $p_i' + p_j' = p_i + p_j$ and $|x_i|/p_i' = |x_j|/p_j'$ can only reduce the maximum. In order for all $|x_k|/p_k$ to be equal it must be that $p_k = |x_k|/\|x\|_1$ for all $j$, in which case $\max_k |x_k|/p_k = \|x\|_1$.

The second claim follows from applying Jensen's inequality to the convex function $x \mapsto x^2$. Specifically, Jensen's inequality shows that for any $p$,
$$\mathbb{E}_{i \sim p}[(|x_i|/p_i)^2] \geqslant \mathbb{E}_{i \sim p}[(|x_i|/p_i)]^2 = \|x\|_1^2$$
This inequality is met for $p_i = |x_i|/\|x\|_1$. $\qquad\square$

To prove Lemma 5.1 we first establish the following.

**Lemma A.2.** *For any matrix $A$ and any probability distribution $p$ on the elements of $A$, we have $|\sigma^2/\tilde{\sigma}^2 - 1| \leqslant \frac{\|A\|_2^2}{\sum_i \|A_{(i)}\|_1^2}$ and $|R/\tilde{R} - 1| \leqslant \frac{\|A\|_2}{\|A\|_1}$.*

*Proof.* Recall that $B_1$ contains one non-zero element $A_{ij}/p_{ij}$, while all its other entries are 0. Therefore, $\mathbb{E}[B_1 B_1^T]$ and $\mathbb{E}[B_1^T B_1]$ are both diagonal matrices where
$$\mathbb{E}[(B_1 B_1^T)_{i,i}] = \sum_j A_{ij}^2/p_{ij} \qquad \text{and} \qquad \mathbb{E}[(B_1^T B_1)_{j,j}] = \sum_i A_{ij}^2/p_{ij} \ .$$
Since the operator norm of a diagonal matrix equals its largest entry we see that
$$\tilde{\sigma}^2 := \max\left\{ \max_i \sum_j A_{ij}^2/p_{ij} \ , \ \max_j \sum_i A_{ij}^2/p_{ij} \right\} = \max\{\|\mathbb{E}[B_1 B_1^T]\|, \|\mathbb{E}[B_1^T B_1]\|\} \ .$$

We will need to bound $\tilde{\sigma}^2$ from below. Trivially, $\tilde{\sigma}^2 \geqslant \|\mathbb{E}[B_1 B_1^T]\| = \max_i \sum_j A_{ij}^2/p_{ij}$. Defining $\rho_i := \sum_j p_{ij}$ and $q_{ij} := p_{ij}/\rho_i$, the second and third inequalities follow from Lemma A.1
$$\tilde{\sigma}^2 \geqslant \max_i \sum_j \frac{A_{ij}^2}{p_{ij}} = \max_i \rho_i^{-1} \sum_j \frac{A_{ij}^2}{q_{ij}} \geqslant \max_i \rho_i^{-1} \|A_{(i)}\|_1^2 \geqslant \sum_i \|A_{(i)}\|_1^2 \ . \tag{5}$$

On the other hand, $\sigma^2 = \max\{\|\mathbb{E}[Z_1 Z_1^T]\|, \|\mathbb{E}[Z_1^T Z_1]\|\}$, where $Z_1 = B_1 - A$. Since $\mathbb{E}[B_1] = A$,
$$\|\mathbb{E}[Z_1 Z_1^T]\| = \|\mathbb{E}[B_1 B_1^T - A B_1^T - B_1 A^T + A A^T]\| = \|\mathbb{E}[B_1 B_1^T] - A A^T\|$$
and, analogously, $\|\mathbb{E}[Z_1^T Z_1]\| = \|\mathbb{E}[B_1^T B_1] - A^T A\|$. Therefore, by the triangle inequality, $|\sigma^2 - \tilde{\sigma}^2| \leqslant \|A\|^2$ and the claim now follows from (5).

Recall that $B_1$ contains one non-zero entry $A_{ij}/p_{ij}$ and that $R = \max \|B_1 - A\|$ over all possible realizations of $p$, i.e., choices of $(i, j)$. Thus, $R = \max \|B_1 - A\| \leqslant \max \|B_1\| + \|A\|$ by the triangle inequality, while if $B_1^* = \arg\max \|B_1\|$, then $R = \max \|B_1 - A\| \geqslant \|B_1^* - A\| \geqslant \|B_1^*\| - \|A\| = \max \|B_1\| - \|A\|$. Since $B_1$ has one non-zero entry, $\max \|B_1\| = \max_{ij} |A_{ij}|/p_{ij} = \tilde{R}$ and, thus, $|R/\tilde{R} - 1| \leqslant \|A\|/\tilde{R}$. Applying Lemma A.1 to $A \in \mathbb{R}^{m \times n}$ with distribution $p$ yields $\tilde{R} \geqslant \|A\|_1$. $\qquad\square$

*Proof of Lemma 5.1.* It suffices to prove that both $|\sigma^2/\tilde{\sigma}^2 - 1|$ and $|R/\tilde{R} - 1|$ are bounded by $1/30$.

Lemma A.2 yields the first inequality below and Condition 2 of Definition 4.1 the second. The third inequality holds for every matrix $A$, with equality occurring when all rows have the same L1 norm.
$$|\sigma^2/\tilde{\sigma}^2 - 1| \leqslant \frac{\|A\|_2^2}{\sum_i \|A_{(i)}\|_1^2} \leqslant \frac{\|A\|_1^2}{30m \sum_i \|A_{(i)}\|_1^2} \leqslant \frac{1}{30} \ .$$
Lemma A.2 yields the first inequality below. The second inequality follows from rearranging the factors in the second inequality above. Condition 3 of Definition 4.1, i.e., $m \geqslant 30$, implies the third.
$$|R/\tilde{R} - 1| \leqslant \frac{\|A\|_2}{\|A\|_1} \leqslant \frac{1}{\sqrt{30m}} \leqslant \frac{1}{30} \ .$$
$\qquad\square$

# B Global minimization over the distribution

To find the probability distribution $p$ that minimizes $\varepsilon_5$ we start by writing $p = \rho_i q_{ij}$, without loss of generality. That is, we decompose $p$ to a distribution $\rho_i \geqslant 0$ over the rows of the matrix, i.e., $\sum_i \rho_i = 1$, and a distribution $q_{ij} \geqslant 0$ within each row $i$, i.e., $\sum_j q_{ij} = 1$, for all $i$. We first prove that (surprisingly) the optimal $q$ has a closed form solution while the optimal $\rho$ is efficiently computable.

For any $\rho$, writing $\varepsilon_5$ in terms of $\rho_i, q_{ij}$ we see that $\varepsilon_5$ is the maximum, over rows $1 \leqslant i \leqslant m$, of

$$\frac{\alpha}{\sqrt{\rho_i}} \sqrt{\sum_j \frac{A_{ij}^2}{q_{ij}} + \frac{\beta}{\rho_i} \max_j \frac{|A_{ij}|}{q_{ij}}} \quad . \tag{6}$$

Observe that since $\rho$ is fixed, the only variables in the above expression for each row $i$ are the $q_{ij}$. Lemma A.1 implies that setting $q_{ij} = |A_{ij}|/\|A_{(i)}\|_1$ simultaneously minimizes both terms in (6). This means that for *every* fixed probability distribution $\rho$, the minimizer of $\varepsilon_5$ satisfies $q_{ij} = \frac{|A_{ij}|}{\|A_{(i)}\|_1}$. Thus, we are left to determine

$$\Phi(\rho) = \max_i \left[ \frac{\alpha\|A_{(i)}\|_1}{\sqrt{\rho_i}} + \frac{\beta\|A_{(i)}\|_1}{\rho_i} \right] \quad .$$

Unlike the intrarow optimization, the two summands in $\Phi$ achieve their respective minima at different distributions $\rho$. To get some insight into the tradeoff, let us first consider the two extreme cases. When $\beta = 0$, minimizing the maximum over $i$ requires equating all $\|A_{(i)}\|_1/\sqrt{\rho_i}$, i.e., $\rho_i \propto \|A_{(i)}\|_1^2$, leading to the distribution we call "row-$L_1$", i.e., $p_{ij} \propto |A_{ij}| \cdot \|A_{(i)}\|_1$. When $\alpha = 0$, equating the $\|A_{(i)}\|_1/\rho_i$ requires $\rho_i \propto \|A_{(i)}\|_1$, leading to the "plain-$L_1$" distribution $p_{ij} \propto |A_{ij}|$.

Nevertheless, since we wish to minimize the maximum over several functions, we can seek $p$ under which all functions are equal, i.e., such that there exists $\zeta > 0$ such that for all $i$,

$$\frac{\alpha\|A_{(i)}\|_1}{\sqrt{\rho_i}} + \frac{\beta\|A_{(i)}\|_1}{\rho_i} = \zeta > 0 \quad .$$

Solving the resulting quadratic equation and selecting for the positive root yields equation (7), i.e.,

$$\rho_i(\zeta) = \left( \frac{\alpha\|A_{(i)}\|_1}{2\zeta} + \sqrt{\left(\frac{\alpha\|A_{(i)}\|_1}{2\zeta}\right)^2 + \frac{\beta\|A_{(i)}\|_1}{\zeta}} \right)^2 \quad . \tag{7}$$

Since the quantities under the square root in (7) are all positive we see that it is always possible to find $\zeta > 0$ such that all equalities hold, and thus (7) does minimize $\varepsilon_5$ for every matrix $A$. Moreover, since the right hand side of (7) is strictly decreasing in $\zeta$, binary search finds the unique value of $\zeta$ such that $\sum \rho_i = 1$.

Finally, recall that our overall goal is to determine the minimizer of $\varepsilon_4 = \max\{\varepsilon_5, \varepsilon_6\}$. We already have determined the minimizer of $\varepsilon_5$. We will now show that for matrices satisfying Condition 1 of Lemma 4.1 the minimizer of $\varepsilon_5$ is also the minimizer of $\varepsilon_4$. We first prove that

**Lemma B.1.** *For any two functions $f, g$, if $x_0 = \arg\min_x f(x)$ and $g(x_0) \leqslant f(x_0)$, then $\min_x \max\{f(x), g(x)\} = f(x_0)$.*

*Proof.*

$$\min_x \max\{f(x), g(x)\} \geqslant \min_x f(x) = f(x_0) = \max\{f(x_0), g(x_0)\} \geqslant \min_x \max\{f(x), g(x)\}$$

□

Thus, it suffices to evaluate $\varepsilon_6$ at the distribution $p$ minimizing $\varepsilon_5$ and check that $\varepsilon_6(p) \leqslant \varepsilon_5(p)$.

We know that $p$ is of the form $p_{ij} = \rho_i|A_{ij}|/\|A_{(i)}\|_1$ for some distribution $\rho$. Substituting this form of $p$ into $\varepsilon_6$ gives (8). Condition 1 of Lemma 4.1, i.e., $\max_j \|A^{(j)}\|_1 \leqslant \min_i \|A_{(i)}\|_1$, allows us to

pass from (9) to (10). Finally, to pass from (10) to (11) we note that the two maximizations over $i$ in (10) involve the same expression, thus externalizing the maximization has no effect.

$$
\varepsilon_6(p) = \max_j \left[ \alpha \left( \sum_i \frac{\|A_{(i)}\|_1 \cdot |A_{ij}|}{\rho_i} \right)^{1/2} + \beta \max_i \frac{\|A_{(i)}\|_1}{\rho_i} \right] \tag{8}
$$

$$
\leqslant \max_j \left[ \alpha \left( \max_i \frac{\|A_{(i)}\|_1}{\rho_i} \cdot \sum_i |A_{ij}| \right)^{1/2} + \beta \max_i \frac{\|A_{(i)}\|_1}{\rho_i} \right]
$$

$$
= \max_j \left[ \alpha \left( \max_i \frac{\|A_{(i)}\|_1}{\rho_i} \cdot \|A^{(j)}\|_1 \right)^{1/2} + \beta \max_i \frac{\|A_{(i)}\|_1}{\rho_i} \right]
$$

$$
\leqslant \alpha \left( \max_i \frac{\|A_{(i)}\|_1}{\rho_i} \cdot \max_j \|A^{(j)}\|_1 \right)^{1/2} + \beta \max_i \frac{\|A_{(i)}\|_1}{\rho_i} \tag{9}
$$

$$
\leqslant \alpha \left( \max_i \frac{\|A_{(i)}\|_1}{\rho_i} \cdot \min_i \|A_{(i)}\|_1 \right)^{1/2} + \beta \max_i \frac{\|A_{(i)}\|_1}{\rho_i} \tag{10}
$$

$$
\leqslant \max_i \left[ \alpha \left( \frac{\|A_{(i)}\|_1}{\rho_i} \cdot \min_i \|A_{(i)}\|_1 \right)^{1/2} + \beta \frac{\|A_{(i)}\|_1}{\rho_i} \right] \tag{11}
$$

$$
\leqslant \max_i \left[ \alpha \frac{\|A_{(i)}\|_1}{\sqrt{\rho_i}} + \beta \max_i \frac{\|A_{(i)}\|_1}{\rho_i} \right]
$$

$$
= \varepsilon_5(p) \ .
$$

## C   Proof of Theorem 4.3

*Proof of Theorem 4.3.* We start by computing the value of $\varepsilon_1$ as a function of $s, \delta$, for the probability distribution $P_0$ minimizing $\varepsilon_5$. Recall that in deriving (7) we established that $\varepsilon_5(P_0) = \zeta_0$, where $\zeta_0$ is such that $\sum_{i=1}^m \rho_i(\zeta_0) = 1$, i.e.,

$$
1 = \sum_{i=1}^m \left( \frac{\alpha\|A_{(i)}\|_1}{2\zeta_0} + \sqrt{\left(\frac{\alpha\|A_{(i)}\|_1}{2\zeta_0}\right)^2 + \frac{\beta\|A_{(i)}\|_1}{\zeta_0}} \right)^2 \leqslant \sum_{i=1}^m \frac{\alpha^2\|A_{(i)}\|_1^2}{\zeta_0^2} + \frac{2\beta\|A_{(i)}\|_1}{\zeta_0} \ . \tag{12}
$$

This yields the following quadratic equation in $\zeta_0$

$$
\zeta_0^2 - \zeta_0 \cdot 2\beta\|A\|_1 - \alpha^2 \sum_i \|A_{(i)}\|_1^2 \leqslant 1 \tag{13}
$$

Treating (13) as an equality and bounding the larger root of the resulting quadratic equation we get

$$
\zeta_0 = O\left( \beta\|A\|_1 + \alpha \sqrt{\sum_i \|A_{(i)}\|_1^2} \right) = O\left( \frac{\log\left(\frac{m+n}{\delta}\right)\|A\|_1}{s} + \sqrt{\frac{\log\left(\frac{m+n}{\delta}\right)\sum_i \|A_{(i)}\|_1^2}{s}} \right) \tag{14}
$$

The second equality is obtain by replacing $\alpha, \beta$ with their corresponding expressions of $\alpha = \sqrt{\log((m+n)/\delta)/s}$ and $\beta = \log((m+n)/\delta)/(3s)$. Recall that to prove Theorem 4.2 we proved that if $A$ meets the conditions of Definition 4.1, then

$$
\min_P \varepsilon_1(p) = \Theta(\zeta_0) \ .
$$

It follows that for $\varepsilon^* = \min_P \varepsilon_1(p)$,

$$
s = O\left( \frac{\log((m+n)/\delta)\sum_i \|A_{(i)}\|_1}{\varepsilon^*} + \frac{\log((m+n)/\delta)\sum_i \|A_{(i)}\|_1^2}{(\varepsilon^*)^2} \right)
$$

The theorem now follows by taking $\varepsilon^* = \varepsilon\|A\|$. $\qquad\qquad\square$

# D    Efficient Parallel Reservoir Sampling

Assume we are to receive a stream of unknown items where the weight of the $i$-th item is $w_i > 0$. We wish to sample a single item from the stream so that each stream item is selected with probability $p_i = w_i/W$, where $W = \sum_i w_i$. Reservoir sampling is the classic solution to this problem: select the very first item in the stream as the "current" sample and from then on have each successive item $i$ replace the current sample with probability $w_i/W_i$, where $W_i = \sum_{j \leqslant i} w_j$.

Assume now that, instead, we wanted to take $s > 1$ items from the stream, but as if the stream was a set and we could sample it *with* replacement. One way to do this is to execute $s$ independent reservoir samplers as above in parallel, as was pointed out in [DZ11]. This, however, requires $\Theta(s)$ active memory and $\Theta(s)$ randomized operations *per item in the stream.*

In forming a sketch matrix $B$, the fact $s = \mathrm{nnz}(B)$ make the above approach impractical, as the overall number of operations required is $\mathrm{nnz}(A)\,\mathrm{nnz}(B) = \Omega(s^2)$. Below we describe an algorithm that requires only $O(\log s)$ *active* memory and $O(1)$ operations per item, instead of $\Theta(s)$ memory and $\Theta(s)$ operations per item, respectively. The first idea is to use the fact that samplers are independent. We can therefore simulate the process above by determining for each item, $a$, the (random) number of samplers, $k$, that would have replaced their current sample with $a$ when it appeared. This random variable is Bernouli distributed and can be sampled efficiently. If this number is greater than zero, we write item $a$ along with $k$ to durable storage (disk) and process the next item in the stream. This processing generates a sketch of the stream on disk, the length of which can be shown to be bounded by $O(s \log(bN))$, where $b = \max_i w_i / \min_i w_i$.

When the stream terminates, we process the sketch from *end to beginning* as follows: for each pair $(a, k)$ we encounter in the sketch we process the $k$ update operations as the throwing of $k$ balls into $s$ bins uniformly at random. This is because, whether item $a$ replaces the current sample, $a'$, of a particular sampler is independent of $a'$. Notice that since we are going over the sketch backwards, the very first ball we place in a bin corresponds to the very last update of the sampler in the original execution. Thus, for each bin, we ignore all but the first ball placement and we stop as soon as each bin has received a ball (thus we also avoid simulating the "irrelevant" part of the naive computation). Performing this simulation only requires a bit-vector of length $s$ in active memory.

Finally, we can avoid even the cost of the bit-vector, as follows. Note that we do not care about the order of the samplers. Only the *number* of samplers that pick any item is important. Therefore, we can simply track the number of empty bins $\ell$ (samplers that are not committed yet) instead of the whole list and update it every time some balls fall into empty bins. The hypergeometric distribution $\mathrm{hypergeometric}(s, \ell, k)$ (see e.g [Ber07] for a more thorough overview) assigns each integer $t$ probability $\binom{\ell}{t}\binom{s-\ell}{k-t}/\binom{s}{k}$. In words, assume we have $s$ bins only $\ell$ of which are empty. If we throw $k$ balls to $k$ different bins uniformly at random, the number of balls that fall in empty bins distributes as $\mathrm{hypergeometric}(s, \ell, k)$.

---

**Input:** An integer $s$ and a stream $(a_1, w_1), (a_2, w_2), ...$
$W \leftarrow 0, \quad T \leftarrow$ empty stack
**for** $(a, w) \in$ the stream **do**
    $W \leftarrow W + w$
    $p = w/W$
    $k = \mathrm{binomial}(s, p)$                    $\rhd$ Number of reservoir samplers that would have picked item $a$.
    **if** $k > 0$ **then**
        Push $(a, k)$ onto $T$
$\ell = s$                              $\rhd$ $\ell$ holds the number of samplers that did not commit on an item yet.
**while** $\ell > 0$ **do**
    $(a, k) = \mathrm{pop}(T)$
    $t = \mathrm{hypergeometric}(s, \ell, k)$
    **if** $t > 0$ **then**                                $\rhd$ $t$ samplers committed to item $a$.
        $\ell = \ell - t$
        **yield:** $(a, t)$

---