[Reviews · NeurIPS 2013]

Submitted by Assigned_Reviewer_6

The paper describes a novel procedure to sample from a sparse data matrix using the spectral norm as a measurement of quality. The procedure is described for the "streaming" setting in which in which a fixed matrix is read from disk and sampled without requiring the entire matrix to reside in memory at any one time.

The paper is overall well written and assumptions and implications are explained clearly. For instance, while the definition of the data matrix in 2.1 appearing to be quite arbitrary at first glance, the authors made a good case that these conditions should hold for most typical inputs. The background sections do a good job of positioning this work in relation to other techniques and the experiments are described clearly and well done. The resultant procedure is quite unique, relying upon an unusually weighted data-dependent sampling procedure and the authors demonstrate that the procedure can match or outperform other matrix sampling methods.

However, the Proof Outline (section 5) is difficult to follow. It is not clear what proof it is trying to outline (for Theorem 4.1?) and it is not clear how Lemma 5.1, 5.2, 5.3 relates, and how the proof outline concludes.


Other issues:
- Equation 7 is written in a rather confusing way. Perhaps split it into two sections: P_ij = \theta |A_{ij}| and \theta = ...
- Page 4, line 210 "matrices B_i". Should this be B^{(l)} ?
- Page 6, line 315. It is not clear where that quadratic equation came from.
- Page 7 line 349. 62 sqrt(2) / 29 ~= 3.02 which is > 3
- The blue "Bernstein" line is completely invisible in many of the plots. Adding some arrows to indicate the location of the Bernstein line will be helpful.
Summary: Few complaints, a well-written paper.

Submitted by Assigned_Reviewer_7

The paper introduces a new sampling method for sampling the entries of large matrices. Unlike deterministic approaches, which only select matrix entries that exceed a given threshold, the authors propose a probabilistic approach for selecting entries of a matrix that are candidates for storage and further analysis. From that perspective, the authors are looking for a matrix B that approximates an input matrix A in an optimal way, where optimality is measured in terms of the loss ||B-A||.

The approach of the authors is supported by precise mathematical arguments, but I was not convinced by this paper. I found the introduction utterly confusing. For example, the simple formula in (1) does not lead to a probability, in the sense that it yields a score between zero and one. Maybe I am totally confused, but I also did not get why B_ij should be in the set (-theta_i, 0, theta_i). I guess it should be in the set (-1/theta_i, 0, 1/theta_i). Moreover, I also did not get why B should be an unbiased estimator of A. The same small mathematical inconsistencies continue to appear in later sections. For example, in Theorem 1.1, the expected value of a matrix is zero. This should probably be the n x m zero matrix?

Most importantly, the motivation for this work was not convincing to me. Why should the machine learning community care about sampling models for matrix data? The authors mention the “streaming model” motivation, where one must decide for every entry whether to keep it or not upon presentation. In such a scenario, probably a deterministic solution based on retaining entries that exceed a threshold would suffice. From that perspective, the last paragraph of Section 3 could not convince me that such a deterministic approach is less useful. I would like to read at least one concrete application where one would need sampling models for matrix data. Probably, such applications exist, but then they should be mentioned in a paper of this kind.
Summary: Interesting problem, but a better justification for this work is needed.

Submitted by Assigned_Reviewer_8

Near-optimal Distribution for Data Matrix Sampling

The paper presents a new sampling methods for generating (random) sparse matrices that best approximate a given matrix in terms of the spectral norm. Several assumptions are made and well motivated, such as the "data matrix" properties and the constraints of the computational model (i.e. streaming data). The exposition is very readable, although some topics such as compressibility and fast indexing are brought up that are then never related back to the method proposed. More to the heart of the matter, the author(s) start with discussing the commonly known L1 and L2 sampling schemes, which serve as reference points throughout the development of the paper and then move on to the matrix Berstein inequality and explain its significance in the context of sparse matrix sampling. A lot of attention is given to proper explanation and intuitive arguments, which makes the paper accessible for non-specialists. The only thing that has to be said is that the text becomes somewhat "talkative" and lengthy, in particular in comparison to the very condensed experimental section. I would recommend shortening these passage, while keeping the gist of the arguments and explanations.

The main contributions of the paper are algorithm 1 together theorem 2.2, which justifies it. I find eq 6 and what feeds into it somewhat hard to digest. Some properties (e.g. certain regimes for s: small, large) are explained in the surrounding text, but it would be nice to find a way of relating this back to the Bernstein inequality that supposedly motivates it. Some of this happens in section 5, but overall the algebraic complexity and notation make this difficult and it would be nice to preserve the crucial link between how even the simplified row-L1 distribution and the Bernstein inequality. With all these caveats around the structure and clarity of the presentation of the main contributions, I deem the result to be significant and the key idea as well as the execution of the proof and algorithmic detail to be highly non-trivial.

The experimental section and the empirical analysis are somewhat disappointing by comparison. First, I cannot even read the graphs in the printed paper (which is a nuisance). Second, there is really no diving deeper on any aspect. Third, no connection to any relevant application is made. Fourth, the spectral norm criteria is thrown out ("it turns out to be more fruitful to consider…") and replaced by a new criterion based on projections to the top singular vectors. The spectral norm objective is not even plotted or mentioned any more. Given how much care has been spent on the theory part, how detailed things have been motivated and then worked through in the proof, this feels like quite a surprise. Shouldn't one go back and change the objective to address the "scaling" issue, for instance?
Summary: Good overview of literature and nice idea to take the Bernstein inequality as a starting point to seek for better sampling distributions. However, in the light of the experimental section and without a somewhat deeper analysis (beyond the proofs), I am not 100% convinced of the superiority of this approach.
Author Feedback

Author rebuttal: General Comment:

We would like to thank the reviewers for helpful comments that will undoubtably help improve our papers. The minor comments are not addressed and will be fixed for the camera ready version

The following is a concise description of our main contribution:
In this paper we provide a helpful method aiding any ML process that relies on spectral properties of a matrix
such as dimension reduction, recommendation, clustering, ranking etc. Specifically, when one wishes to perform one of these tasks
on a huge matrix, (s)he will have to work with a smaller sketch of the matrix that best describes the original.
If the matrix arrives in a streaming fashion, we offer the best provable (and empirically supported) method of creating the sketch (which is also optimal under reasonable assumptions).
To the best of our knowledge, there is no known superior method even when it is allowed to perform O(1) passes over the data


Our work could thus be extremely helpful in the ML community for anyone required to perform a `spectrally motivated' process on a large data set.
Regarding the novelty of our paper:
1) We are the first to provide a method with some assurance compared to the optimal.
2) We are the first to provide experimental results on real data matrices. These show that indeed our method is superior to others.


++++++++++++ Reviewer 1 +++++++++++++

>> "the Proof Outline (section 5) is difficult to follow. ... and it is not clear how Lemma 5.1, 5.2, 5.3 relates, and how the proof outline concludes."

We agree with the reviewer, yet the argument is inherently intricate. We struggled to simplify it as much as we could, the result of which was presented.
Currently, we devote the first two sentences of Section 5 to say in words what we do: "In what follows... bring us closer to a closed form solution." In the camera-ready version we will expand to provide the reader with a more explicit road-map of how the different pieces of the proof relate to each other.



++++++++++++ Reviewer 2 +++++++++++++

>> the simple formula in (1) does not lead to a probability.
Setting theta_i as done later in the paper makes sure it is a proper distribution.

>> I guess it should be in the set (-1/theta_i, 0, 1/theta_i).
You are 100% right! This is a glitch that will be fixed in the camera ready version. Notice that the essential point remains: to represent the sketch, we only need to record a single real number (theta_i) for each row and only the sign of each retained entry, i.e., a single bit.


>> Moreover, I also did not get why B should be an unbiased estimator of A.
Using an unbiased estimator is not necessary but is a very sensical approach that lets us employ strong concentration of measure inequalities.
Note that, like us, all previous work on the subject use unbiased estimators (precluding tiny matrix entries that can safely be ignored)

>> the expected value of a matrix is zero. This should probably be the n x m zero matrix?
Correct. This is an abbreviated notation. If you found it confusing, so will others. We will change for the camera ready.

>> Why should the machine learning community care about sampling models for matrix data?
Please see the general comments.


>> In such a scenario, probably a deterministic solution based on retaining entries that exceed a threshold would suffice.
While this intuitive idea sometimes works in practice, retaining large entries can be shown not to work in general or produce much larger sketches.

++++++++++++ Reviewer 3 +++++++++++++

>> the text becomes somewhat "talkative" and lengthy, in particular in comparison to the very condensed experimental section.
We tried to make the paper as readable as possible to non experts.
That, by lengthening the introduction on body at the expense of the experimental and proofs sections.
We fully acknowledge that expert readers would prefer a different balance (so would we if the situation was reversed).
We will shift some of the weight from the intro to the experiment section and proofs in the camera ready version.

>> I find eq 6 and what feeds into it somewhat hard to digest.
We will simplify for the camera ready.


>> I cannot even read the graphs in the printed paper.
We will make the plots more readable.

>> no connection to any relevant application is made.
The entire paper stayed away from specific applications for two reasons.
First, the method is general and really does not depend on the type of data matrix.
Second, the space limitation prevented this. In fact, even the presented experiments were hard to fit in.

>> the spectral norm criteria is thrown out
Note that the measure we end up using for the experiments is a natural generalization of the spectral norm criterion. At the same time, optimizing directly for that measure is very cumbersome. The key point demonstrated by the experiments is the following. Since we work with data matrices that are already quite sparse (the hardest case), the number of samples required to achieve non-trivial approximations w.r.t. to the spectral norm requires a very large number of samples. In the case of our sparsest matrix (Enron), as many as in the original matrix. By switching to the projection-measure, we can demonstrate that even for sparse matrices and even when subsampling aggressively, our sketches provide a good estimate of the original matrix. We will describe this phenomenon more accurately in the camera ready version.